# Transcriptome analysis of *Tamarix ramosissima* leaves in response to NaCl stress

**Yahui Chen[1,2], Guangyu Wang[2¤a], Hongxia Zhang[3¤b], Ning Zhang**  **[2]\*, Jiang Jiang[1¤c]\*, Zhizhong Song[3]\***

1 Collaborative Innovation Center of Sustainable Forestry in Southern China of Jiangsu Province, Nanjing Forestry University, Nanjing, Jiangsu, China, 2 Department of Forest Resources Management, University of British Columbia, Vancouver, British Columbia, Canada, 3 The Engineering Research Institute of Agriculture and Forestry, Ludong University, Yantai, Shandong, China

¤a Current address: Department of Forest Resources Management, Forest Sciences Centre, University of British Columbia, Vancouver, British Columbia, Canada
¤b Current address: College of Agriculture, Ludong University, Yantai, Shandong, China
¤c Current address: Faculty of Foresty, Nanjing Forestry University, Nanjing, Jiangsu, China
* ecologyjiang@gmail.com (JJ); zhangningswing@outlook.com (NZ); 3614@ldu.edu.cn (ZS)

**Data Availability Statement:** All relevant data are within the paper and its Supporting information files.

**Funding:** This work was jointly supported by the following grants: the Agricultural Variety

## Abstract

Halophyte *Tamarix ramosissima*. Lcdcb (*T. ramosissima*) are known as the representative of *Tamarix* plants that are widely planted in salinized soil. However, molecular mechanisms towards salt tolerance and adaptation are largely rare. In this study, we carried out RNA-sequence and transcriptome analysis of *T. ramosissima* in response to NaCl stress, screened differentially expressed genes (DEGs) and further verified by qRT-PCR. Results showed that 105702 unigenes were spliced from the raw data of transcriptome sequencing, where 54238 unigenes were retrieved from KEGG, KOG, NR, and SwissProt. After 48 hours of NaCl treatment, the expression levels of 6374 genes were increased, and 5380 genes were decreased in leaves. After 168 hours, the expression levels of 3837 genes were up-regulated and 7808 genes were down-regulated. In particular, 8 transcription factors annotated to the KEGG Pathway were obtained, involing the WRKY and bZIP transcription family. In addition, KEGG pathway annotation showed that expression of 39 genes involved in ROS scavenging mechanisms were significantly changed, in which 21 genes were up-regulated and 18 genes were down-regulated after 48 hours as well as 15 genes were up-regulated and 24 genes were down-regulated after 168h. Simultaneously, the enzyme activities of SOD and POD were significantly enhanced under NaCl treatment, but the enzyme activity of CAT was not significantly enhanced. Moreover, WRKY, MYB and bZIP may participate in the process of salt resistance in *T. ramosissima*. This study provides gene resources and a theoretical basis for further molecular mechanisms of salt tolerance in *T. ramosissima*.

Improvement Project of Shandong Province (2019LZGC009), China's National Science Foundation through grants (32071612), and the China Scholarship Council (202108320311). Agricultural Variety Improvement Project of Shandong Province provided experimental materials (Hongxia Zhang). China's National Science Foundation through grants (Jiang Jiang) and the China Scholarship Council provide test sites and data collection(Yahui Chen). Agricultural Variety Improvement Project of Shandong Province, China's National Science Foundation through grants and the China Scholarship Council provided great help in research design, data collection and analysis.

**Competing interests:** NO authors have competing interests.

## 1. Introduction

Salinized soil has high salt content and poor soil physical and chemical properties, which seriously hindered the growth and development of plants [1]. Salinized soil contains a lot of Na$^+$ type salt which can destroy the stability of protein and membrane, and produces osmotic stress and ion poisoning to initiate reactive oxygen ROS (reactive oxygen species) signals in the cell, make dysfunction of the cell, affect the growth of plants, and causes plant death in severe cases [2]. In the past decades, the area of salinized soil has continued to expand due to global human activities and climate changes. Favorably, it is becoming a hotspot to carry out afforestation, restore salinized soil and improve the ecological environment in salinized soil areas.

In recent years, RNA-Seq technology has been widely used to study molecular mechanisms of plant resistance to adverse stresses, including salt stress [3, 4]. Previous studies showed that reactive oxygen species (ROS) would be produced in multiple cell compartments, including chloroplasts [5], mitochondria [6], and peroxisomes [7] under salt stress conditions. A relatively low concentration of ROS is known as an important signal molecule to regulate the normal plant growth and responses to abiotic stresses [8, 9]. However, excessive accumulation of ROS can adversely cause cell oxidative damage [10]. To adapt to ROS damage, higher plants have evolved corresponding regulatory mechanisms to maintain the stability of life activities. Notably, ROS scavenging enzymatic systems, such as superoxide dismutase (SOD), peroxidase (POD), catalase (CAT), ascorbate peroxidase (APX), and glutathione peroxidase (GPX), are prone to play important roles to adapt to undesired abiotic stresses [11]. In addition, plants can also take advantage of the ascorbic acid- reduced glutathione (AsA-GSH) cycle, which contains APX, glutathione reductase (GR) and AsA, and GSH, and so on, to eliminate the damage of ROS [12].

In particular, transcription factors are the most important regulators for plants to respond to various abiotic stresses [13]. In details, reports are focused on the involvement of transcription factors of WRKY [14], bHLH [15], bZIP [16] NAC [17], MYB [18], AP2/ERF [19] under salt stress. In cotton, the WRKY transcription factor gene *GhWRKY34* was induced by salt stress in transgenic *Arabidopsis* [20]. In *Paspalum notatum*, the WRKY transcription factor genes are involved in regulating the expression of SOD and related oxidoreductase genes to adapt to salt stress [21]. Notably, *Tamarix* plants have evolved a complex regulatory network for a long time to adapt to the adverse abiotic stresses [22]. *T. hispida* is often used as a biological material to explore the molecular mechanisms towards salt stress tolerance. Overexpression of the *T. hispida ThCOL2* gene can regulate the activity of protective enzymes and reduce the accumulation of O$^{2-}$ and H$_2$O$_2$ that enhanced the ROS scavenging ability and improved the adaptability of transgenic *T. hispida* under salt stress [23]. Overexpression of *ThbZIP1* enhanced the activity of POD and SOD, increased the content of soluble sugar and soluble protein that further improved salt tolerance in transgenic *T. hispida* [24]. *T. ramosissima* usually grows in arid and semi-arid regions with high salt content [25]. Moreover, low concentration (<100mM) NaCl promoted while high concentration (≥200mM) NaCl stress inhibited the growth of *T. Ramosissima* [26]. Liu and her colleagues found that *T. ramosissima* exhibited the strongest salt tolerance among 3 *Tamarix* species, including *Tamarix gansuensis* H.Z.Zhang, *Tamarix leptostachys* Bunge, and *Tamarix ramosissima*. Lcdcb, and *T. ramosissima* were chosen as the representative species of *Tamarix* plants for further mechanisms studies.

In this study, we performed high-throughput transcriptome sequencing in *T. ramosissima* under NaCl stress and screened and verified DEGS at the transcriptional level. This study lays a theoretical basis to reveal molecular mechanisms towards salt tolerance in *Tamarix* plants, and provides gene resources for further variety breeding of salt-tolerant *Tamarix* plants.

## 2. Materials and methods

### 2.1. Plant materials

*T. ramosissima* seedlings were provided by the Dongying Experimental Station of Shandong Academy of Forestry Sciences. Experiments were completed at the Key Laboratory of Forest Tree Genetic Breeding and Biotechnology of the Ministry of Education of Nanjing Forestry University from October 2019 to March 2021. 5-month-old *T. ramosissima* seedlings with uniform growth were transferred to a 24-well hydroponic box (size: 40cm*30cm*16cm), supplemented with 1/2 Hoagland nutrient solution, and then placed in a greenhouse that was maintained at 26 ± 2˚C (day) whose relative humidity stays between 40% and 55% for 1 month after training before treatment. The culture solution was changed every 3rd day.

### 2.2 NaCl treatment

In the control group (CK), seedlings were suffered with 1/2 Hoagland nutrient solution. In the treatment group, seedlings were cultured in 1/2 Hoagland nutrient solution, supplemented with 200 mM NaCl. 8 plants were used in each group, and the experiments were repeated 3 times. The culture solution was changed every 3rd day. The leaf samples were collected at 0h, 48h, and 168h, respectively, and immediately frozen in liquid nitrogen, and then moved to a -80˚C refrigerator for storage.

### 2.3 Phenotype and antioxidative enzyme activity analysis in *T. ramosissima* leaves

Leaves of *T. ramosissima* were collected after 0h, 48h and 168h of NaCl treatment, respectively, and the distribution of salt secretion on the leaf surface was observed using a JSZ6S stereo microscope (Jiangnan, China). The activities of SOD [27], POD [28] and CAT [29] were determined and analyzed, according to the description of the commercial Extraction Kits (Jiancheng Limit Co., Nanjing, China).

### 2.4 Transcriptome sequencing

The frozen leaf samples were used for 3-generation high-throughput transcriptome sequencing, using Illumina HiSeq™4000, in Guangzhou GENE Denovo Company. The purified PCR products were analyzed by pair-end sequencing (PE150) on the platform according to standard operations, and then fastp was used for quality control [30]. The original data was first filtered to obtain clean reads, then assembled [31]. These assembled fragments without N terminal obtained by reading overlap are used as the assembled Unigene, and then use Blast2 GO [32] and KOBAS [33] to obtain Gene Ontology (GO) function and Kyoto Encyclopedia of Genes and Genomes (KEGG) annotation. The Illumina raw sequencing data were submitted to the National Center for Biotechnology Information (NCBI) Short Reads Archive (SRA) database under accession number SRP356215.

### 2.5 Screening methods for differentially expressed genes

DESeq2 software was used to analyze the reads count data to obtain the final correct FDR value (FDR value means BH corrected *p* value) [34]. A corrected *p* value of <0.05 is considered to be significantly enriched. Based on the results of the difference analysis, genes with FDR value <0.05 and |log2FC| > 1 are considered to be significantly different genes.

## 2.6 Quantitative Real-Time PCR (qRT-PCR) validation of DEGs

Eight putative genes (*Unigene0104732*, *Unigene0028215*, *Unigene0083695*, *Unigene0069097*, *Unigene0090596*, *Unigene0024962*, *Unigene0007135* and *Unigene0088781*) were randomly selected to verify the accuracy of the RNA-Seq results by the qRT-PCR technique. The total RNA was extracted with Omega RNA Extraction Kit (Shanghai, China), and then reverse-transcribed the 1-strand cDNA by using the PrimerScript™ RT Master Mix Kit (TaKaRa, Dalian, China). Specific primers of DEGs are designed via the Primer-BLAST server (S1 Table). qRT-PCR samples were labelled with PowerUp™ SYBR Green Master mix reagent (Thermo Fisher, China) and then performed on ABI ViiA™ 7 Real-time PCR system (ABI, USA). A total of 3 biological repeats were performed, each with 4 technical repetitions. *Actin* was used as the internal reference gene, and the relative expression level was calculated by the $2^{-\Delta\Delta Ct}$ method.

## 3 Results

### 3.1 Phenotype and antioxidative enzyme activity analysis in *T. ramosissima* leaves

In the CK group, there was no salt secretion in the leaves of *T. ramosissima* at 0h, 48h and 168h. However, leaves began to excrete a small amount of salt at 48h under 200 mM NaCl treatment, and the salt secretion reached the maximum amount at 168h under NaCl treatment (Fig 1). These findings showed that the amount of salt secretion in leaves increased along with the prolongation of NaCl treatment time.

The activities of SOD, POD and CAT in the leaves of *T. ramosissima* showed an increasing trend under 200mM NaCl for 48h and 168h, compared with the control group. In particular, the activities of SOD and POD were significantly higher than those of the control group after either 48h or 168h. The CAT activity increased slightly but had no significant change, compared with the control group (Fig 2). These results showed that SOD and POD activities under salt stress in the leaves of *T. ramosissima*.

### 3.2 Sequencing quality analysis

Using IlluminaHiSeq™4000, obtained multiple high-quality bases at 0h, 48h and 168h were obtained in *T. ramosissima* leaves under 200mM NaCl stress, and Q20 reached more than 95%, and Q30 reached more than 90% and the GC content is above 44% (Table 1), indicating that the quality of the transcriptome sequencing is relatively high, which is reliable for further analysis.

### 3.3 Unigene basic notes

Results showed that a total of 105,702 Unigenes were spliced in this study. There are 53,385, 46062, 31587, and 36087 Unigenes with gene annotations in the Nr, KEGG, KOG, and SwissProt databases, respectively, and a total of 27,670 Unigenes were simultaneously screened with annotations in the four major databases (Fig 3).

### 3.4 Quantitative expression analysis of DEGs

Taking the CK group as the control, the transcription data of *T. ramosissima* under NaCl stress treatment at 0h, 48h and 168h were compared and analyzed, respectively. DEGs were screened with the standard of FDR value <0.05, *p* value <0.05 and |log2FC| > 1 after correction. In the CK-0h v.s. NaCl-48h comparison group, a total of 11,754 gene expression levels were detected, in which 6374 genes were up-regulated and 5380 genes were down-regulated under NaCl

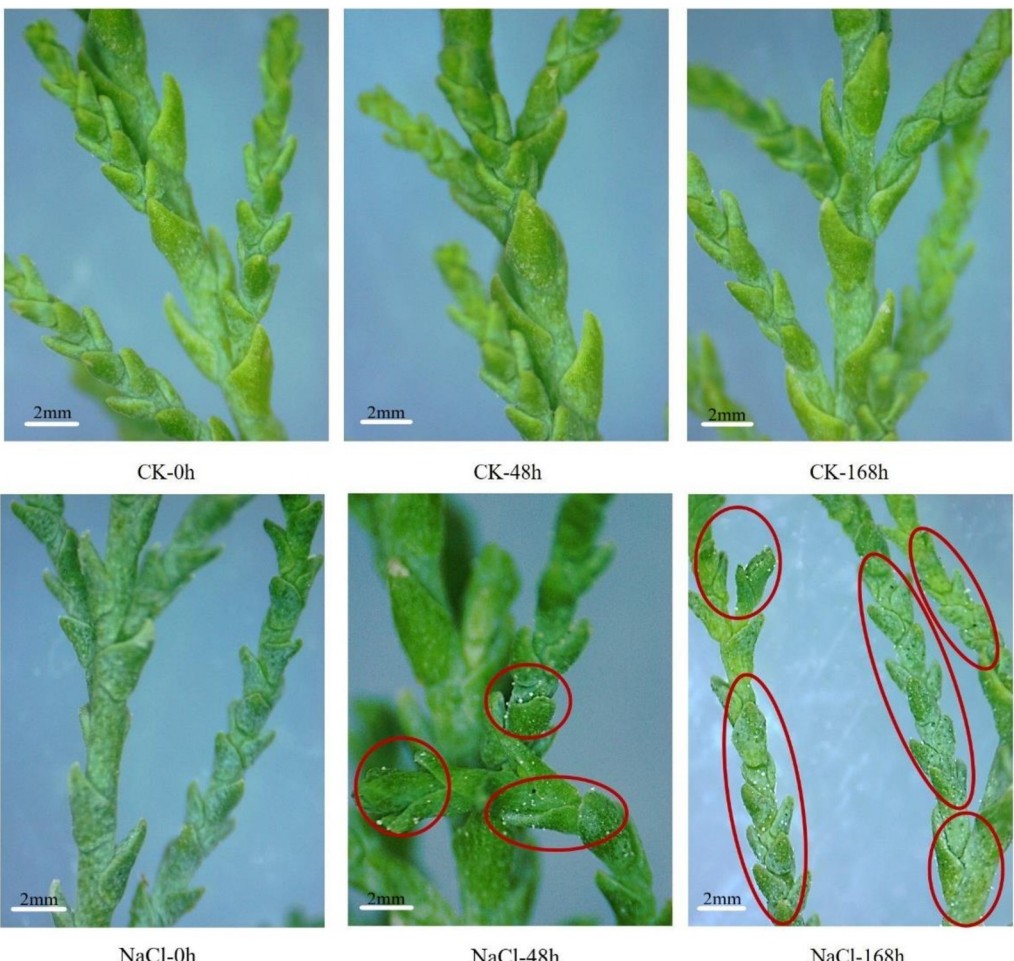

**Fig 1. Salt secretion from *T. ramosissima* leaves under NaCl stress.** [Leaf salt secretion are checked in the leaves of *T. ramosissima* at 0h, 48h and 168h under 200mM NaCl treatment (The red labels indicate the salt secretion products)].

treatment. In the CK-0h v.s. NaCl-168h comparison group, a total of 7768 gene expression changes were detected, in which 2542 genes were induced and 5226 genes were reduced under NaCl treatment (Fig 4).

## 3.5 GO analysis of DEGs

Through GO annotation analysis, the above-mentioned DEGs can be divided into 3 categories: biological processes, cellular components, and molecular functions, and a total of 51 different classification groups were observed (Fig 5). In detail, in the CK-0h v.s. NaCl-48h comparison group, the up-regulated genes were slightly more than the down-regulated genes, while in the CK-0h v.s. NaCl-168h comparison group, the down-regulated genes were significantly more than the up-regulated genes. In the broad category of biological process, DEGs are mainly concentrated in cellular process, metabolic process, single-organism process and response to stimulus (Table 2). In the molecular function category, DEGs are mainly concentrated in catalytic activity, binding, transporter activity and structural molecule activity (Table 3). Among the major categories of cell components, DEGs are mainly enriched in cell, cell part, organelle and membrane (Table 4). In addition, the number of up-regulated DEGs was significantly lower

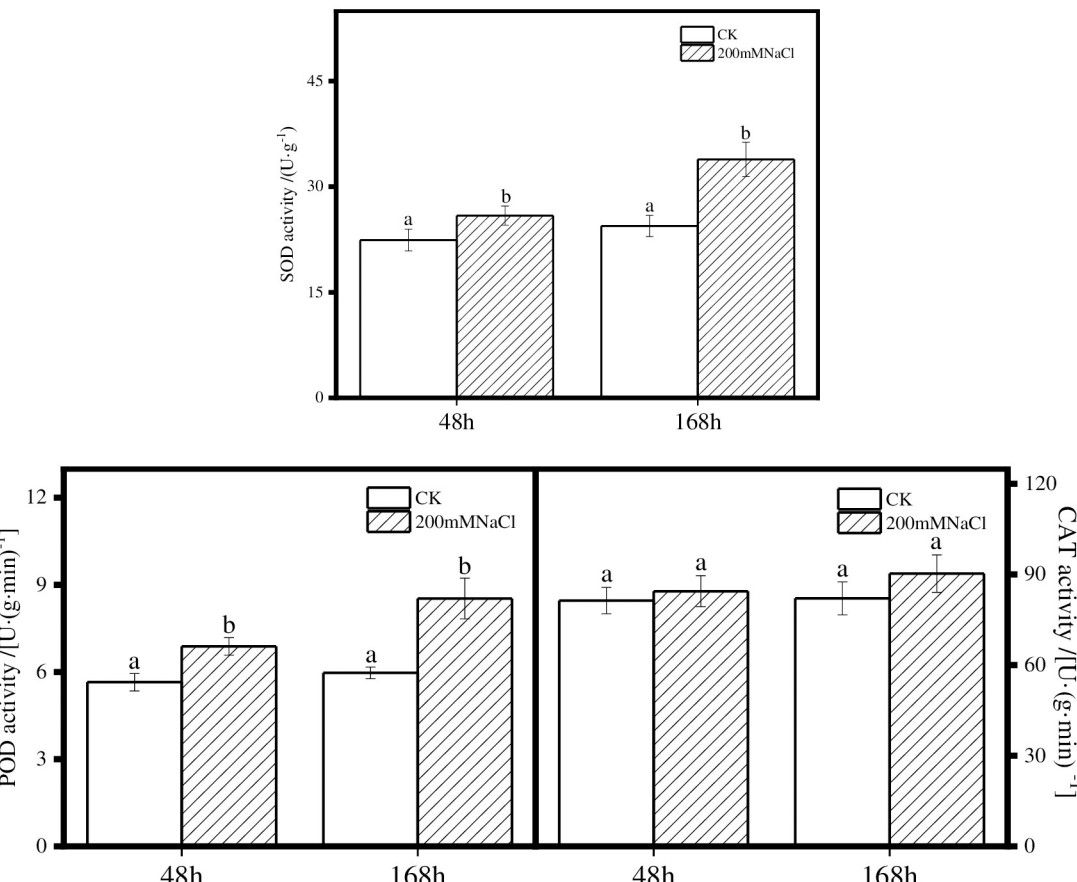

**Fig 2. Changes of SOD, POD and CAT enzyme activities in leaves of *T. ramosissima* under salt stress.** (Note: The different letters indicate significant differences at the *p* value < 0.05. The graph illustrates the changes of enzyme activities in SOD, POD and CAT compared with CK group at 48h and 168h after 200mM NaCl treatment).

and the overall number of DEGs decreased, along with the time of NaCl treatment. We speculate that *T. ramosissima* leaves may respond to high NaCl stress by affecting the expression level of related DEGs at the transcription level.

3.6 KEGG pathway analysis of DEGs in *T. ramosissima* leaves under NaCl stress KEGG pathway analysis showed that 1762 and 1366 DEGs were annotated in the comparison group

**Table 1. Filtered reads quality statistics.**

| Sample | Raw data (bp) | Clean data (bp) | Q20 (%) | Q30 (%) | GC (%) |
| --- | --- | --- | --- | --- | --- |
| CK1-0h | 6341519400 | 6017997220 | 97.34% | 92.65% | 45.15% |
| CK2-0h | 6216526200 | 6057604903 | 97.54% | 93.11% | 45.12% |
| CK3-0h | 6627399900 | 6507140412 | 97.77% | 93.55% | 45.24% |
| NaCl1-48h | 6654895800 | 6541177895 | 98.78% | 95.93% | 45.04% |
| NaCl2-48h | 6888168900 | 6782061623 | 98.72% | 95.71% | 44.94% |
| NaCl3-48h | 6720560700 | 6605897734 | 98.79% | 95.94% | 44.90% |
| NaCl1-168h | 6691086000 | 6551036697 | 98.85% | 96.17% | 45.42% |
| NaCl2-168h | 6181114500 | 6032346218 | 98.83% | 96.16% | 45.44% |
| NaCl3-168h | 6396633600 | 6256461214 | 98.93% | 96.43% | 45.35% |

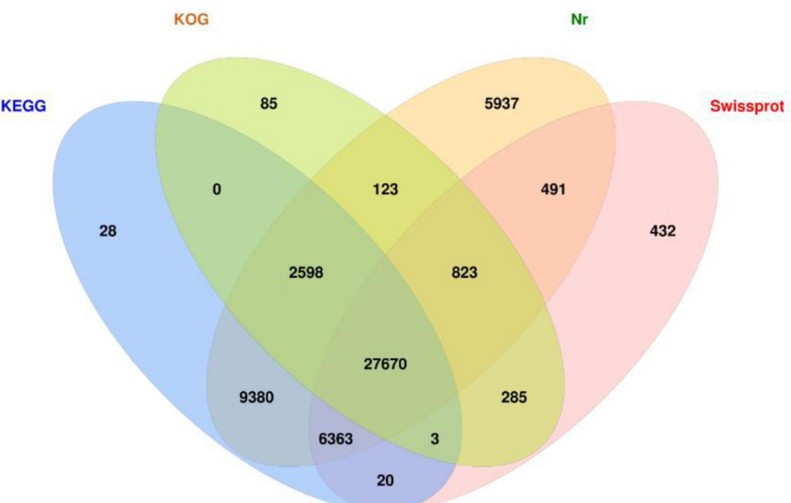

**Fig 3. Annotation diagrams obtained form 4 major databases.** (Distribution map of 4 major databases annotated to genes).

of CK-0h v.s. NaCl-48h and CK-0h v.s. NaCl-168h, respectively (Fig 6), which directly reflected the changes of gene expression in leaves of *T. ramosissima* under NaCl stress. Among the top 10 KEGG pathways from the CK-0h v.s. NaCl-48h comparison group, Ribosome (ko03010) annotated to 435 DEGs, accounting for 24.69%, followed by biosynthesis of secondary metabolites (ko01110), phytopathogen interaction (ko04626), phenylpropane biosynthesis (ko00940), plant hormone signal transduction (ko04075) and plant MAPK signaling pathway (ko04016), respectively, annotated to 416 (23.61%), 88 (4.99%), 69 (3.92%), 69 (3.92%) and 59

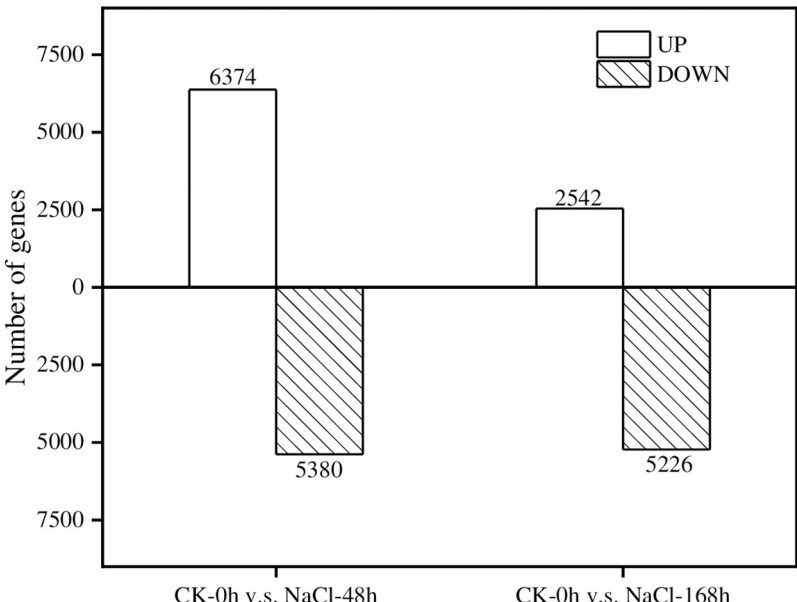

**Fig 4. Analysis of DEGs.** (The differentially expressed genes were up-regulated and down-regulated in the comparison groups of CK-0h v.s. NaCl-48h and CK-0h v.s. NaCl-168h).

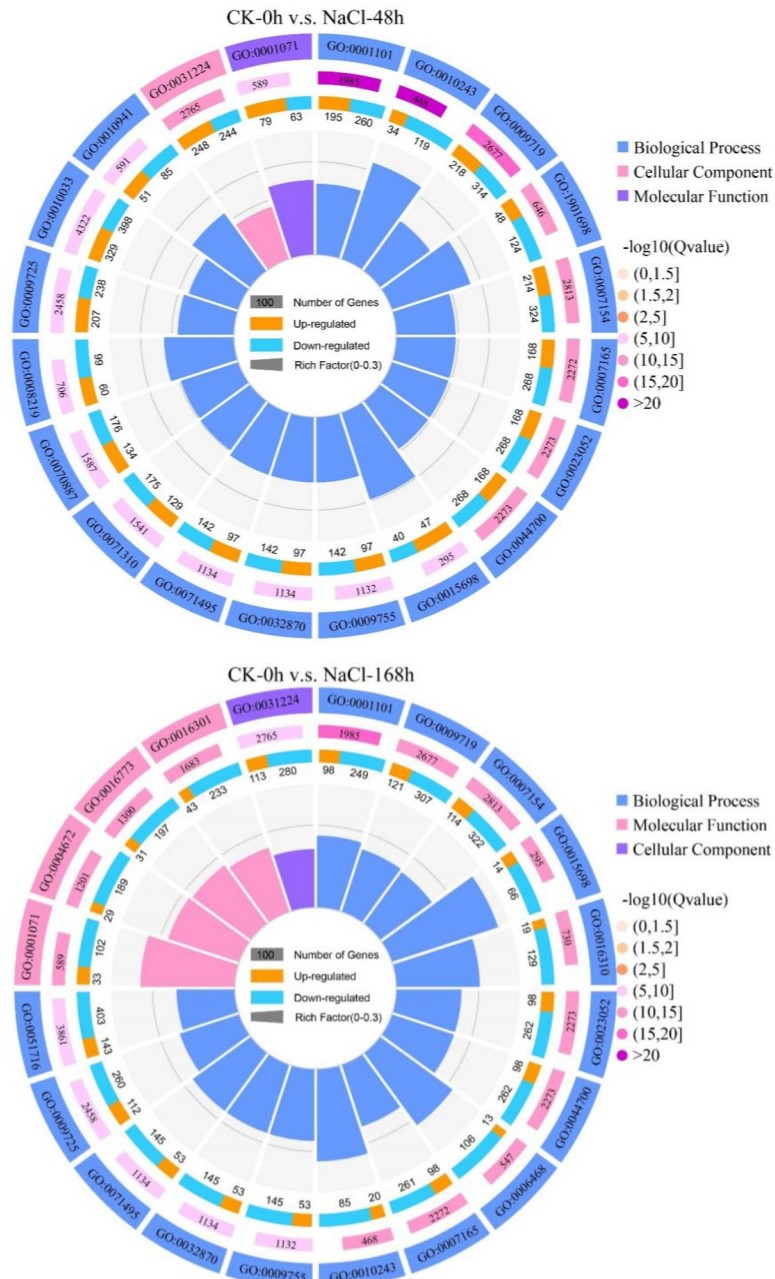

**Fig 5. GO enrichment analysis of DEGs.** (The first and outer circle: the top 20 GO terms are enriched, outside the circle is the scale of the number of genes. Different colors represent different Ontologies. The second circle: the number of the GO term in the background gene and the Q value. The more genes, the better the bars. Long, the smaller the Q value, the darker the color. The third circle: the bar graph of the ratio of up-regulated genes, the dark color represents the proportion of up-regulated genes, and the light color represents the proportion of down-regulated genes. The specific value is displayed below. The fourth and inner circle: the ratio of each GO term Rich Factor value (number of differential genes in this GO term divided by all numbers), background grid lines, each grid represents 0.1).

**Table 2. GO enrichment analysis of the biological process of DEGs.**

| GO ID | GO Term | CK-0hv.s.NaCl-48h (DEGs) | | CK-0hv.s.NaCl-168h (DEGs) | |
|---|---|---|---|---|---|
| | | Up | Down | Up | Down |
| GO:0023052 | signaling | 168 | 268 | 98 | 262 |
| GO:0065007 | biological regulation | 659 | 563 | 269 | 728 |
| GO:0050789 | regulation of biological process | 584 | 514 | 240 | 655 |
| GO:0002376 | immune system process | 73 | 124 | 29 | 130 |
| GO:0032501 | multicellular organismal process | 398 | 276 | 150 | 394 |
| GO:0050896 | response to stimulus | 796 | 875 | 411 | 962 |
| GO:0051704 | multi-organism process | 178 | 242 | 95 | 261 |
| GO:0032502 | developmental process | 519 | 373 | 201 | 511 |
| GO:0044699 | single-organism process | 1269 | 1222 | 592 | 1400 |
| GO:0048511 | rhythmic process | 18 | 15 | 15 | 13 |
| GO:0048519 | negative regulation of biological process | 90 | 37 | 22 | 70 |
| GO:0048518 | positive regulation of biological process | 48 | 61 | 23 | 75 |
| GO:0000003 | reproduction | 222 | 156 | 93 | 226 |
| GO:0040011 | locomotion | 1 | 4 | 1 | 0 |
| GO:0022414 | reproductive process | 220 | 152 | 91 | 223 |
| GO:0098754 | detoxification | 4 | 0 | 1 | 2 |
| GO:0051179 | localization | 457 | 436 | 197 | 498 |
| GO:0001906 | cell killing | 0 | 1 | 0 | 4 |
| GO:0040007 | growth | 83 | 68 | 46 | 102 |
| GO:0022610 | biological adhesion | 2 | 4 | 4 | 10 |
| GO:0009987 | cellular process | 1395 | 1456 | 659 | 1638 |
| GO:0008152 | metabolic process | 1420 | 1479 | 659 | 1630 |
| GO:0071840 | cellular component organization or biogenesis | 499 | 417 | 217 | 495 |

3.35% DEGs. Among the top 20 pathways from the CK-0h v.s. NaCl-168h comparison group, the metabolic pathway (ko1100) was annotated to 614 DEGs, accounting for 44.95%, followed by the Biosynthesis of secondary metabolites (ko01110), Ribosomes (ko03010), Oxidative phosphorylation (ko00190), Plant hormone signal transduction (ko04075) and Plant pathogen interaction (ko04626), respectively, annotated to 351 (25.70%), 296 (21.67%), 104(7.61%), 72

**Table 3. GO enrichment analysis of molecular function of DEGs.**

| GO ID | GO Term | CK-0hv.s.NaCl-48h (DEGs) | | CK-0hv.s.NaCl-168h (DEGs) | |
|---|---|---|---|---|---|
| | | Up | Down | Up | Down |
| GO:0001071 | nucleic acid binding transcription factor activity | 79 | 63 | 33 | 102 |
| GO:0004871 | signal transducer activity | 11 | 26 | 3 | 30 |
| GO:0005215 | transporter activity | 146 | 117 | 57 | 152 |
| GO:0016209 | antioxidant activity | 12 | 23 | 6 | 18 |
| GO:0009055 | electron carrier activity | 6 | 9 | 5 | 7 |
| GO:0060089 | molecular transducer activity | 14 | 19 | 4 | 18 |
| GO:0005198 | structural molecule activity | 119 | 165 | 52 | 130 |
| GO:0003824 | catalytic activity | 999 | 1062 | 458 | 1212 |
| GO:0098772 | molecular function regulator | 16 | 8 | 8 | 17 |
| GO:0000988 | transcription factor activity, protein binding | 1 | 1 | 1 | 2 |
| GO:0005488 | binding | 939 | 967 | 415 | 1104 |
| GO:0045182 | translation regulator activity | 0 | 0 | 0 | 0 |

**Table 4. GO enrichment analysis of cellular component of DEGs.**

| GO ID | GO Term | CK-0hv.s.NaCl-48h (DEGs) | | CK-0hv.s.NaCl-168h (DEGs) | |
|---|---|---|---|---|---|
| | | Up | Down | Up | Down |
| GO:0044425 | membrane part | 311 | 296 | 146 | 333 |
| GO:0016020 | membrane | 686 | 732 | 342 | 789 |
| GO:0031012 | extracellular matrix | 1 | 2 | 0 | 2 |
| GO:0044421 | extracellular region part | 3 | 3 | 8 | 2 |
| GO:0044420 | extracellular matrix component | 0 | 1 | 0 | 0 |
| GO:0005576 | extracellular region | 76 | 97 | 49 | 112 |
| GO:0009295 | nucleoid | 2 | 2 | 0 | 1 |
| GO:0030054 | cell junction | 193 | 226 | 93 | 248 |
| GO:0099512 | supramolecular fiber | 2 | 0 | 0 | 5 |
| GO:0019012 | virion | 0 | 3 | 0 | 2 |
| GO:0044423 | virion part | 0 | 3 | 0 | 2 |
| GO:0031974 | membrane-enclosed lumen | 6 | 12 | 8 | 9 |
| GO:0044422 | organelle part | 502 | 506 | 232 | 497 |
| GO:0032991 | macromolecular complex | 327 | 311 | 119 | 298 |
| GO:0043226 | organelle | 1167 | 1077 | 595 | 1180 |
| GO:0005623 | cell | 1345 | 1290 | 667 | 1430 |
| GO:0044464 | cell part | 1340 | 1284 | 664 | 1428 |

(5.27%) and 63 (4.61%) DEGs. In summary, DEGs are significantly enriched on the KEGG pathway, such as metabolic pathways, biosynthesis of secondary metabolites, and plant hormone signal transduction, in *T. ramosissima* under NaCl treatment.

## 3.6 Analysis of antioxidant DEGs in *T. ramosissima* leaves under NaCl stress

Results showed that the expression of 39 ROS scavenging-related DEGs in the leaves of *T. ramosissima* were changed under salt treatment (Table 5). Notably, there were 21 up-regulated genes and 18 down-regulated genes from the CK-0h v.s. NaCl-48h comparison group. The largest number of up-regulated genes is GST (7), followed by POD (4), CAT (4), APX (3), GR (2) and SOD (1). The largest number of down-regulated genes is GST (8), followed by POD (4), SOD (3), APX (2) and GPX (1). In the CK-0h v.s. NaCl-168h comparison group, there were 15 up-regulated genes and 24 down-regulated genes. In particular, the largest number of up-regulated genes are CAT(3), APX(3) and GST(3), followed by SOD(2), POD(2), and GR (2). The largest number of down-regulated genes is GST(12), followed by POD(6), SOD(2), APX(2), CAT(1) and GPX(1) (Fig 7). These results showed implying that the antioxidant mechanism might be initially enhanced in *T. ramosissima* leaves accompanied with corresponding physiological responses to resist NaCl stress during the first 48 hours. However, the antioxidant mechanism might be initially inhibited in *T. ramosissima* leaves under a long time (168 h) of NaCl treatment, just to adapt to salt stress.

## 3.7 Analysis of transcription factor DEGs in *T. ramosissima* leaves under NaCl stress

Transcription factors play a major role in regulating plant growth and adaptation to adverse environments, including salt stress. In this study, According to the transcriptome sequencing of *T. ramosissima leaves*, many transcription factors were discovered. In this study, we did

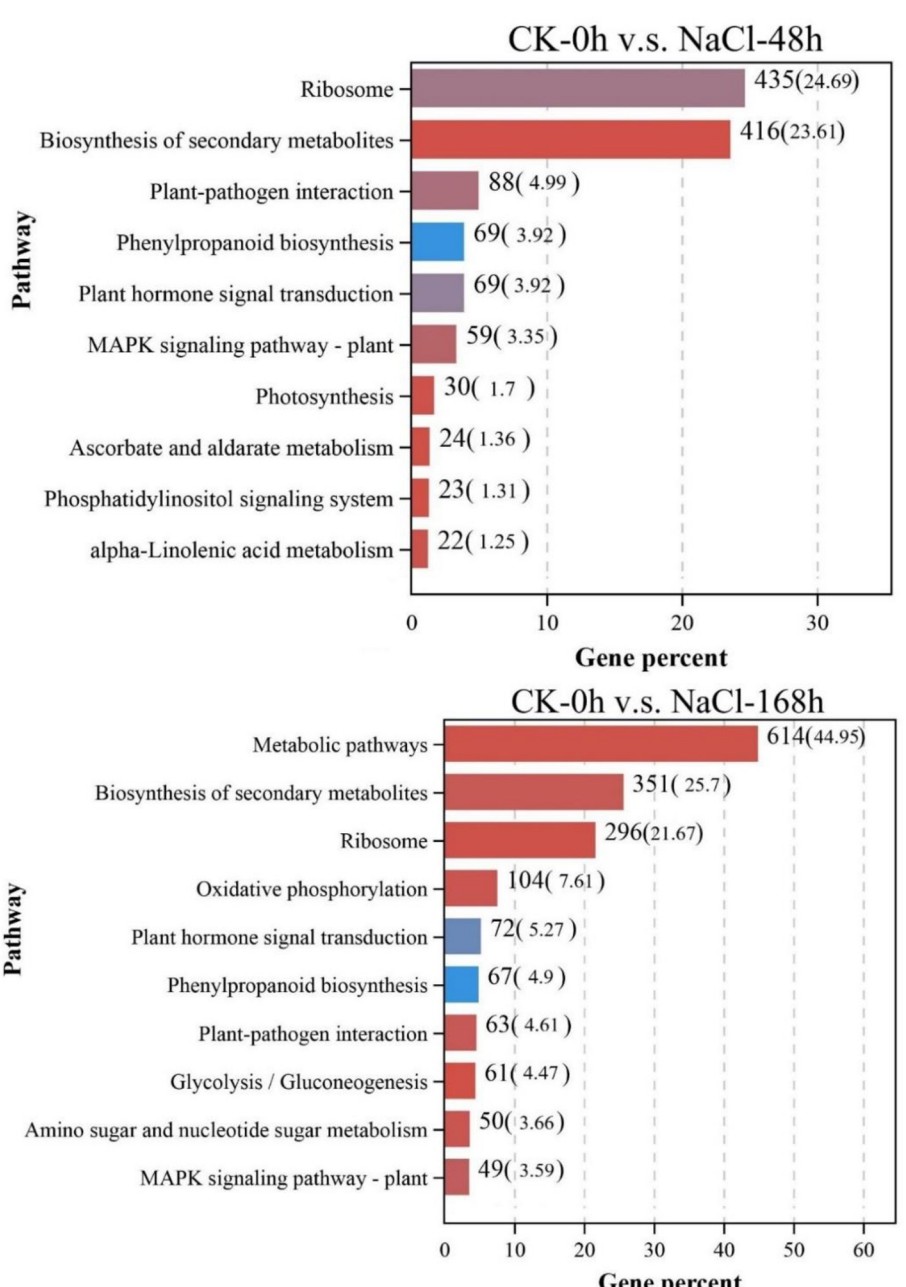

**Fig 6. Top 10 pathway analysis.** (Distribution of differentially expressed genes in the top 10 KEGG pathway in the CK-0hv.s.NaCl-48h and CK-0hv.s.NaCl-168h comparison groups).

select 8 statically significant DEGs, which were specifically observed and annotated in the KEGG database. In particular, 5 WRKY (ko04626 and ko04016) annotated genes were responsive to NaCl treatment. In particular, the expression levels of *Unigene0010090*, *Unigene0077293* and *Unigene0079542* exhibited a downward trend at 0h, 48h and then an upward trend at 168h. The expression level of *Unigene0014406* and *Unigene0024962* continuously increased until 168h. In addition, 3 bZIP transcription factors were annotated to the KEGG pathway (ko04016, ko01100, ko01110, ko01200, ko01212, ko04146, ko00071, ko00640,

**Table 5. Antioxidant DEGs annotated to the KEGG pathway.**

| Pathway | Gene ID | Description | Log$_2$FoldChange | |
|---|---|---|---|---|
| | | | CK-0hv.s. NaCl-48h | CK-0hv.s. NaCl-168h |
| SOD | | | | |
| ko04146 | Unigene0033269 | SOD4 protein, partial | -1.01 | -0.17 |
| | Unigene0049419 | superoxide dismutase [Mn] | 7.82 | 9.54 |
| | Unigene0050462 | superoxide dismutase | -1.51 | -0.60 |
| | Unigene0082550 | superoxide dismutase | -0.57 | 0.26 |
| POD | | | | |
| ko01100;ko01110;ko00940 | Unigene0009260 | peroxidase 20 | -0.46 | -0.59 |
| | Unigene0013825 | peroxidase | 1.79 | -1.76 |
| | Unigene0013827 | peroxidase | 1.17 | -0.61 |
| | Unigene0014843 | peroxidase | -2.95 | -1.80 |
| | Unigene0029752 | peroxidase 17 | 1.51 | -0.44 |
| | Unigene0049353 | peroxidase 5 | -4.98 | 0.70 |
| | Unigene0086491 | peroxidase 52 | 0.85 | -0.74 |
| | Unigene0094375 | peroxidase 31 | -0.32 | 1.59 |
| CAT | | | | |
| ko01100;ko01110;ko01200; ko00630;ko04146;ko04016; ko00380 | Unigene0046159 | catalase isozyme 1 | 0.58 | 0.73 |
| | Unigene0046160 | catalase, partial | 0.77 | 1.93 |
| | Unigene0087092 | leaf catalase | 0.07 | -0.48 |
| | Unigene0103080 | catalase isozyme 1 | 5.74 | 12.41 |
| APX | | | | |
| ko01100;ko00480 | Unigene0008032 | L-ascorbate peroxidase 3 | -0.45 | -0.24 |
| | Unigene0008033 | L-ascorbate peroxidase 3 | 0.49 | 0.60 |
| | Unigene0008513 | peroxidase domain-containing | 0.59 | -0.51 |
| | Unigene0048033 | cytosolic ascorbate peroxidase | -0.02 | 0.08 |
| | Unigene0105664 | thylakoid ascorbate peroxidase precursor, partial | 0.90 | 1.55 |
| GPX | | | | |
| ko01100;ko0048; ko00590 | Unigene0035407 | glutathione peroxidase | -0.18 | -0.65 |
| GST | | | | |

(*Continued*)

**Table 5.** (Continued)

| Pathway | Gene ID | Description | Log$_2$FoldChange | |
|---|---|---|---|---|
| | | | CK-0hv.s. NaCl-48h | CK-0hv.s. NaCl-168h |
| ko01100;ko00480 | *Unigene0001041* | glutathione S-transferase | -3.17 | -11.56 |
| | *Unigene0004890* | glutathione S-transferase T1-like | -1.88 | -0.68 |
| | *Unigene0007072* | glutathione S-transferase U17-like | -0.08 | 0.49 |
| | *Unigene0012650* | glutathione S-transferase Mu 1-like | 13.69 | 7.20 |
| | *Unigene0015109* | glutathione S-transferase U8-like | 0.11 | -0.42 |
| | *Unigene0020552* | glutathione S-transferase | -0.14 | -0.07 |
| | *Unigene0041633* | microsomal glutathione S-transferase 3-like | 0.08 | -0.34 |
| | *Unigene0048538* | glutathione S-transferase U10-like | -3.33 | -2.73 |
| | *Unigene0056773* | glutathione S-transferase | -0.70 | 0.14 |
| | *Unigene0064942* | glutathione S-transferase L3 | 0.28 | -0.10 |
| | *Unigene0069058* | glutathione-S-transferase | 0.33 | 0.92 |
| | *Unigene0069060* | glutathione S-transferase L3-like | -0.03 | -2.16 |
| | *Unigene0081745* | glutathione S-transferase U10-like | 0.05 | -0.14 |
| | *Unigene0082147* | glutathione S-transferase F11-like | 2.47 | -0.45 |
| | *Unigene0098941* | glutathione S-transferase U9 | -0.17 | -5.91 |
| GR | | | | |
| ko01100;ko00480 | *Unigene0075696* | glutathione reductase | 0.47 | 0.12 |
| | *Unigene0098587* | glutathione reductase-like | 8.98 | 10.52 |

ko00410, ko01040, ko00592 and ko04075). The expression level of *Unigene0026888* and *Unigene0008868* exhibited a downward trend at 48h and then an upward trend at 168h, while *Unigene0010561* showed a downward trend at 48h and then an upward trend at 168h (Table 6).

### 3.8 Quantitative Real-Time PCR (qRT-PCR) validation of differential expression

We further randomly selected 8 DEGs involved in salt stress for qRT-PCR verification (S2 Table). Results showed that the expression level of *Unigene0104732*, *Unigene0083695* and *Unigene0069097* were induced at 48h but reduced at 168h, while genes of *Unigene0090596*, *Unigene0007135* and *Unigene0088781* were decreased at 48h but increased at 168h. Notably, *Unigene0024962* was continuously increased while *Unigene0028215* was continuously decreased under NaCl stress. These qRT-PCR verification results are completely consistent with the expression trends observed from the transcriptome sequencing analysis (S1 Fig). Nonetheless, the transcriptome data obtained in this study is accurate and reliable.

## 4. Discussion

*Tamarix* plants have evolved a series of complex mechanisms to resist and adapt to salt stress in the long term. In particular, *Tamarix* plants have typical multicellular salt glands, which can

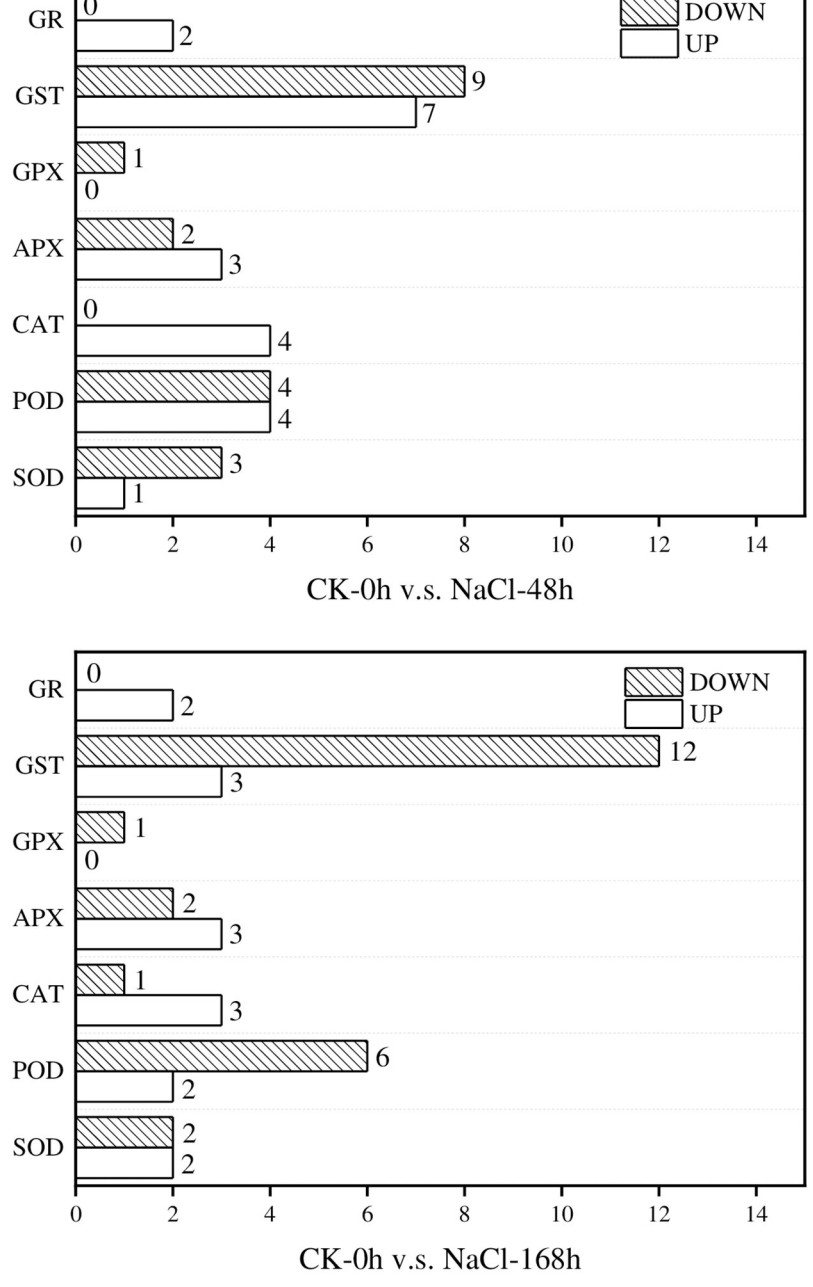

**Fig 7. Antioxidant mechanism related genes under NaCl treatment in *T. ramosissima*.** (Notes: GR, glutathione reductase; GST, glutathione S-transferases; APX, ascorbate peroxidase; GPX, glutathione peroxidase; CAT, Catalases; POD, peroxidase; SOD, Superoxide dismutase. The number of up-regulated and down-regulated activities of each enzyme in the reactive oxygen species scavenging mechanism shown in the figure in the comparison of CK V.S. NaCl-48h and CK V.S. NaCl-168h).

reduce plant damage from ion poisoning, which is one of the important morphological characteristics of *Tamarix* plants to adapt to the saline environment [25]. In this study, electron microscopy analysis showed that leaf salt secretion increased with the prolongation of NaCl treatment, suggesting that *T. tamariska* may alleviate and respond to the toxicity of salt stress via the salt secretion pathway, thereby adapting to the adverse long-term salt stresses.

**Table 6. Gene annotation of transcription factors.**

| Gene ID | Description | Pathway | Log$_2$ fold change | |
|---|---|---|---|---|
| | | | CK-0hv.s.NaCl-48h | CK-0hv.s.NaCl-168h |
| *Unigene0010090* | Transcription factor WRKY33 | ko04626;ko04016 | -0.75 | -0.40 |
| *Unigene0014406* | WRKY DNA-binding protein 27 | ko04626;ko04016 | 0.46 | 0.79 |
| *Unigene0024962* | WRKY transcription factor 1 | ko04626 | 0.64 | 0.69 |
| *Unigene0077293* | WRKY transcription factor | ko04626;ko04016 | -1.96 | -2.04 |
| *Unigene0079542* | WRKY transcription factor 11 | ko04626 | -0.20. | 0.18 |
| *Unigene0026888* | bZIP4 | ko04016 | 0.67 | 0.36 |
| *Unigene0008868* | bZIP2 | ko01100;ko01110;ko01200;ko01212;ko04146;ko00071;ko00640;ko00410; ko01040;ko00592 | 0.6439 | -0.011 |
| *Unigene0010561* | bZIP10 | ko04075 | -0.8186 | -0.38502 |

The response of plants under salt stress is quite complex that needs multiple gene families and regulatory mechanisms involved in many biological pathways, including metabolism, signal transduction, energy production and transportation, ion penetration and transportation [35]. A comprehensive transcriptome analysis of *T. ramosissima* plants helps to reveal the molecular mechanisms of *Tamarix* plants in response to salt stress.

## 4.1 Enhanced active oxygen scavenging capacity in *T. ramosissima*

Plants are usually affected by various unfavorable environmental factors, including salt stress, that results in a large amount of ROS accumulation. In this present study, 39 ROS scavenging-related DEGs were significantly regulated under NaCl stress, implying that ROS scavenging mechanisms were indispensable for *T. ramosissima* plants under NaCl stress.

SOD, POD and CAT enzymes are involved in ROS scavenging in plant cells. Under abiotic stresses, expression of SOD, POD and CAT related genes are prone to be up-regulated [36]. Notably, the expression level of the *TaSOD1.7* gene in leaves increased significantly, and the salt tolerance of transgenic wheat was enhanced [37]. In this study, 1 SOD, 4 POD and 4 CAT related genes were induced, while 3 SOD and 4 POD related genes are down-regulated at 48 h of NaCl treatment. Even168 hours after finishing NaCl treatment, there are still 2 SOD, 2 POD and 3 CAT related genes were up-regulated and 2 SOD, 6 POD and 1 CAT related genes were down-regulated. Together, these genes mentioned above may contribute to the increased enzyme activities of SOD and POD in *T. ramosissima* leaves under NaCl treatment.

Both APX and GPX are the key enzymes for enzymatically removing ROS, protecting cells by catalyzing H$_2$O$_2$, and playing an important role in plant adversity stress. APX family genes are involved in plant tolerance to drought, heat, salt, oxidation and biological stresses [38]. In this study, the expression of 3 APX related genes was up-regulated and 2 genes were down-regulated under NaCl treatment. However, only one GPX related gene was down-regulated. We guess that *APX* and *GPX* genes in *T. ramosissima* leaves may respond to salt stress in different ways.

Glutathione S-transferase (GST) is involved in detoxification and antioxidant defence, protecting plants from different abiotic stresses and adversities, and playing multiple roles in plant growth and development [39–42]. In particular, GST was involved in salt stress and GST-related genes are non-sensitive to low and medium NaCl (≤100 mM) concentration but are sensitive to high NaCl (≥200 mM) stress [39, 46]. In this study, 3 GST related genes were up-

regulated and 8 genes were down-regulated in the leaves at 48 h, and 3 genes were up-regulated and 12 genes were down-regulated at 168 h, implying that there may exist both positive and negative regulation of GST in *T. ramosissima* under NaCl stress, and the negative regulation may be the dominant.

GR is one of the plant antioxidant enzymes and a flavoprotein oxidoreductase, which exists in eukaryotes and prokaryotes, and plays an important role in the elimination of ROS in the process of plant oxidative stress [43]. The expression of *Oryza sativa GR3* [44] and *Jatropha JcGR* [45] are all up-regulated under salt stress. Similar to the previous studies, two GR related genes were up-regulated in *T. ramosissima* leaves under NaCl stress. We speculate that GR might be involved in the adaptation of *T. ramosissima* to NaCl stress in a positive direction.

## 4.2 Responsive transcription factors under NaCl treatment ins *T. ramosissima*

Transcription factors are indispensable for plants to respond to abiotic stresses [13]. WRKY is one of the most important transcription factor families, which has been verified to participate in a variety of metabolic processes and plays an important role in the regulation of transcriptional reprogramming related to plant biological and abiotic stress responses [46–48]. In this study, *Unigene0024962*, which encodes a WRKY transcription factor, was up-regulated by NaCl treatment, while *Unigene0079542* was decreased at 48 h but increased at 168 h, which is consistent with the previous reports in *WRKY* family genes in *soybean* [49] and *Reaumuria trigyna* [50]. The expression of *CaWRKY27* in pepper was down-regulated by NaCl stress [51], which is in contrast to our findings. Notably, *Unigene0014406* was continuously induced by NaCl treatment, suggesting that this gene may be inevitably involved in the response of *T. ramosissima* to salt stress. In addition, bZIP transcription factors play an important regulatory role in plant growth and environmental stress response [52, 53]. The expression of *Unigene0008868* was significantly up-regulated by NaCl treatment, which is consistent with that of Huang's findings in ramie, implying that the bZIP transcription factor plays an important regulatory role in response to salt stress [54]. Moreover, MYB is one of the most diversified transcription factor families in plants in terms of quantity and function, which plays important roles under abiotic stresses [55, 56]. In this study, the expression level of *Unigene0088781* decreased in the first 48 hours and then increased in a long time under NaCl treatment, suggesting that this gene may be active, especially during a specific period under salt stress. Similar findings were also observed in *Medicago sativa* seedlings [57] and *Rosa rugosa* petals [58].

## 5. Conclusions

The raw data of transcriptome sequencing of *T. ramosissima* leaves in response to NaCl stress was spliced into 105702 Unigenes, and 54238 annotated Unigenes were retrieved in the 4 major functional databases of KEGG, KOG, NR and SwissProt. The expression profiles of DEGs are slightly different between short time (48h) and long time (168) treatments. In particular, ROS scavenging genes and transcription factor encoding genes (including WRKY, MYB and bZIP family) are sensitive to NaCl treatment with distinct regulatory statuses. This study provides the theoretical basis and gene resource for further molecular mechanisms towards salt tolerance in *T. ramosissima*.

## Supporting information

**S1 Fig. Verification of DEGs by qRT-PCR.**
(PDF)

**S1 Data.**
(XLSX)

**S1 Table. Sequences of specific primers.**
(PDF)

**S2 Table. Randomly select 8 differentially expressed genes.**
(PDF)

## Acknowledgments

This work was jointly supported by the following grants: the Agricultural Variety Improvement Project of Shandong Province (2019LZGC009).

## Author Contributions

**Conceptualization:** Yahui Chen, Hongxia Zhang.

**Data curation:** Yahui Chen, Zhizhong Song.

**Formal analysis:** Guangyu Wang, Jiang Jiang.

**Funding acquisition:** Guangyu Wang.

**Investigation:** Yahui Chen, Guangyu Wang, Zhizhong Song.

**Methodology:** Guangyu Wang, Jiang Jiang.

**Project administration:** Yahui Chen.

**Resources:** Guangyu Wang, Hongxia Zhang, Jiang Jiang, Zhizhong Song.

**Software:** Yahui Chen.

**Validation:** Hongxia Zhang, Jiang Jiang.

**Visualization:** Yahui Chen, Zhizhong Song.

**Writing – original draft:** Yahui Chen.

**Writing – review & editing:** Ning Zhang.

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
