## [Decision Letter · Decision Letter 0]

25 Jan 2022

PONE-D-21-38993Transcriptome analysis of Tamarix ramosissima leaves in response to NaCl stressPLOS ONE

Dear Dr. Zhang,

Thank you for submitting your manuscript to PLOS ONE. After careful consideration, we feel that it has merit but does not fully meet PLOS ONE’s publication criteria as it currently stands. Therefore, we invite you to submit a revised version of the manuscript that addresses all points raised by reviewers.

We look forward to receiving your revised manuscript.

Kind regards,

Guangxiao Yang, Ph.D

Academic Editor

PLOS ONE

Journal Requirements:

"NO"

"We are grateful to Guangyu Wang Professor and his group in the University of British Columbia Forest Science Center for collaborative studies. This work was jointly supported by the following grants: the Agricultural Variety Improvement Project of Shandong Province (2019LZGC009), the National Natural Science Foundations of China (32071612), and the China Scholarship Council (202108320311)."

"NO"

5. Please upload a copy of Supporting Information Figure 1 and Table 1 and 2 which you refer to in your text on page 22.

Reviewers' comments:

Reviewer's Responses to Questions

**Comments to the Author**

1. Is the manuscript technically sound, and do the data support the conclusions?

Reviewer #1: Partly

Reviewer #2: Partly

2. Has the statistical analysis been performed appropriately and rigorously? 

Reviewer #1: I Don't Know

Reviewer #2: Yes

3. Have the authors made all data underlying the findings in their manuscript fully available?

Reviewer #1: Yes

Reviewer #2: Yes

4. Is the manuscript presented in an intelligible fashion and written in standard English?

Reviewer #1: No

Reviewer #2: No

5. Review Comments to the Author

Reviewer #1: Abstract

1. L26-34: reorganize and summarize the core contents.

Method

2.1 and 2.2：Whether the seedlings, Hoagland nutrient solution, and treatment conditions are sterile? If not, the nutrient medium is easy to be polluted, resulting in bacterial treatment of seedlings.

L96 format error: temperature description

Result

1. Why is the treatment group designed for 2d (48h) and 7d (168h)? The response at the transcriptome level is usually relatively rapid. Did such a long time of NaCl treatment produce phenotypic differences of T. ramosissima? IF it is, pictures should be provided,

2. 3.3 The title is inappropriate.

3. 3.4 What is the significance of the analysis of GO notes？Figure2 and Table 2,3,4…Go enrichment analysis can be performed instead of GO annotation analysis statistics…..And what led the authors to this conclusion about “T. ramosissima leaves may respond to high NaCl stress by inhibiting the expression level of related DEGs at the transcription level”?

4. 3.6 According to what the 39 genes were concerned and screened from all DEGs? the GO annotation or KEGG or others

5. What led the authors to this conclusion about “antioxidant mechanism might be initially inhibited in T. ramosissima leaves under long time (168 h) of NaCl treatment, just to adapt to salt stress”? The results are not enough to support the conclusion….. The two diagrams in Figure 5 can be combined.

6. 3.7 The title is inappropriate. According to what the 8 genes were concerned and screened from all DEGs? Only these eight? Are there other TFs involved in stress regulation.

7. 3.8 Unigenes mentioned in 3.6 and 3.7 should be selected and analyzed by qRT-PCR.

8 The authors should detect the related enzyme activities such as POD, SOD, CAT etc. from control and treated T. ramosissima to support the conclusion.

Discussion

1. The author should condense the discussion part.

2. 4.1 L284: The results are not enough to support the conclusion.

L290: “Overexpression of SbGST in transgenic tobacco plays a crucial role in salt stress tolerance” What is the role?

4.2

L310-313:What do the authors want to say?

L322-323: What led the authors to this conclusion about “that this gene needs to be active especially in a specific period under salt stress”? The results are not enough to support the conclusion

Others

1. Where's the legends of the figures？

2. Writing problems in L149, 152, 160, 210,223,225,255,264,271

Reviewer #2: A transcriptome analysis of T. ramosissima in response to NaCl stress was performed in this study, and some differentially expressed genes (DEGs) were found. However, the methods, results and conclusions should be considered more appropriately and rigorously.

Here are the details:

Introduction Section:

Lines 65-66: GhWRKY34 is not a wheat gene, please correct it after carefully reading the reference 20.

Materials and Methods Section:

1. In terms of the materials, why choose leaves not roots to study molecular mechanisms towards salt stress?

2. Lines 109-110: The filtering parameters of Fastp should be added. The original data was first filtered to obtain clean reads, then assembled! Not “the original data was filtered and assembled to obtain clean reads.” Please correct it, and add the method as well as software used in assembly.

Results section:

Table 1 : Please use scientific notation for the data.

Line 170: “We speculate that T. ramosissima leaves may respond to high NaCl stress by inhibiting the expression level of related DEGs at the transcription level.” What exactly does these down-regulated “DEGs” related to?

Line 201-202: “Results showed the expression of 39 ROS scavenging-related DEGs in the leaves of T.ramosissima were changed under salt treatment.” Please revise it.

Lines 223-224: How many transcription factors in DEGs? Why only WRKY and bZIP were mentioned? What’s the P value and log2FC of these 8 genes?

Line 237-238: What kind of genes are Unigene0104732, Unigene0083695 and Unigene0069097? Why randomly choose these three genes instead of choosing genes mentioned above, like WKRY, bZIP?

One of the main conclusion,“T. ramosissima may adapt to salt stress by enhancing the ROS scavenging mechanism, including SOD, POD, CAT, APX, GPX, GST, and GR.” I don’t think this conclusion is reliable. As the result showed there were some SOD, POD and CAT down-regulated whether after 48 h or 168 h salt treatment. The enzyme activity of SOD, POD, CAT should be measured to see whether ROS scavenging mechanism is enhanced.

Besides, the article needs careful revision; there are many typos and format error in it.

6. PLOS authors have the option to publish the peer review history of their article (what does this mean?). If published, this will include your full peer review and any attached files.

Reviewer #1: No

Reviewer #2: No

---

## [Author Response · Author response to Decision Letter 0]

2 Feb 2022

Journal: PLOS ONE

Title: Transcriptome analysis of Tamarix ramosissima leaves in response to NaCl stress

Manuscript ID: PONE-D-21-38993

Dear Dr. Guangxiao Yang, 

Thank you for giving us the opportunity to resubmit our revised paper for publication in Plos One. We greatly appreciate the efforts of you, the associate editor and all reviewers for the constructive comments that have helped shape this manuscript into a better form. We have thoroughly revised the article and addressed all the concerns by either editing the manuscript or clarifying the details in the revised manuscript. Please see the detailed point-by-point response (marked in blue) added below. A 'point-by-point response' to each of the reviewer comments and a marked-up version of the manuscript with the changes highlighted in blue were also uploaded with the clean version of the manuscript files.

Sincerely,

Ning Zhang, Ph.D.

Faculty of Forestry, The University of British Columbia, 

Vancouver, British Columbia, Canada 

 

Additional requirements

Response: Thanks for your reminder. The manuscript has been revised to meet PLOS ONE's style requirement.

Response: The Funding Information has been revised.

("The authors gratefully acknowledge financial support from the Agricultural Variety Improvement Project of Shandong Province 2019LZGC009, China’s National Science Foundation through grants 32071612 and the China Scholarship Council 202108320311.") We note that you have provided funding information that is not currently declared in your Funding Statement. However, funding information should not appear in the Acknowledgments section or other areas of your manuscript. We will only publish funding information present in the Funding Statement section of the online submission form.

("The authors gratefully acknowledge financial support from the Agricultural Variety Improvement Project of Shandong Province 2019LZGC009, China’s National Science Foundation through grants 32071612 and the China Scholarship Council 202108320311.")

Response: The Funding Information has been revised in the Funding Statement．

Response:

Data Availability Statement

All data that support the findings of this study are available. The Illumina raw sequencing data were submitted to the National Center for Biotechnology Information (NCBI) Short Reads Archive (SRA) database under accession number SRP356215.

 

Response to Reviewer 1

Dear Reviewer:

Many thanks for all your valuable comments and suggestions. We carefully revised the manuscript and tried our best to improve it to be a better state to meet the journal’s requirements. The revised manuscript has been submitted to the journal, and looking forward to your further consideration. 

Abstract 

1. Lines:26-34: reorganize and summarize the core contents.

Response: According to your suggestion, we reorganized the abstract and improved greatly in the revised manuscript. Again, thank you for your suggestions. 

Method：

2. 2.1 and 2.2：Whether the seedlings, Hoagland nutrient solution, and treatment conditions are sterile? If not, the nutrient medium is easy to be polluted, resulting in bacterial treatment of seedlings. 

Response: Thanks a lot for your valuable comments. To be honest, the Hoagland nutrient solution used in the study was sterilized before treatment. Before NaCl treatment, the cuttings seedlings were rinsed 3-5 times with sterilized distilled water, and then transferred to the incubator, which was disinfected with 75% alcohol before treatment. Notably, the nutrient solution was changed every third day to avoid media pollution, especially under 26 ℃ in the short-term treatment.

3.Line96 format error: temperature description.line 96 format error: temperature description： “laced in……at 26 (± 2) ℃ .”

Response: Sorry for the incorrect format. It has been revised in the manuscript. 

Results:

1. Why is the treatment group designed for 2d (48h) and 7d (168h)? The response at the transcriptome level is usually relatively rapid. Did such a long time of NaCl treatment produce phenotypic differences of T. ramosissima? IF it is, pictures should be provided.

Response: The purpose of selecting 2d (48h) and 7d (168h) is to observe the changes of gene expression under salt treatment in both the short and the long term. According to previous reports (Song and Su, 2013, Weed Sci; Song et al., 2015, Bio Plantaraum; Song et al., 2016, Sci Horticulturae; Liang et al., 2020 Biomed Res Int; Chen et al., 2021, Int J Genomics), 2d (48h) after stress treatment is a typical time point to monitor the expression levels of responsive genes. While under 7d (168h) treatment, plants exhibited significant physiological phenotype and it is better to discover which DEGs are responsive under salt stress.

2. 3.3 DEGs quantitative analysis

Response: Thank you for your useful suggestions. In the revised manuscript, it was changed into ‘Quantitative expression analysis of DEGs’.

3. 3.4 What is the significance of the analysis of GO notes？Figure2 and Table 2,3,4…Go enrichment analysis can be performed instead of GO annotation analysis statistics…..And what led the authors to this conclusion about “T. ramosissima leaves may respond to high NaCl stress by inhibiting the expression level of related DEGs at the transcription level”?

Response: Appreciate your valuable suggestions, and definitely, Go enrichment analysis was performed instead of GO annotation analysis statistics in the revised submission. GO analysis can clearly understand the expression levels of DEGs in three categories: biological processes, cellular components, and molecular functions. According to the distribution of the subclasses of DEGs in each subclass under each major class, it is easy to find which subclasses of DEGs’ expression levels were sensitive to salt stress.

4. 3.6 According to what the 39 genes were concerned and screened from all DEGs? the GO annotation or KEGG or others.

Response: Among all these DEGs, only 39 ROS-related DEGs were specifically found and annotated in the KEGG database.

5. What led the authors to this conclusion about “antioxidant mechanism might be initially inhibited in T. ramosissima leaves under long time (168 h) of NaCl treatment, just to adapt to salt stress”? The results are not enough to support the conclusion….. The two diagrams in Figure 5 can be combined.

Response: Frankly speaking, we did analyze the enzymatic activities of SOD, POD and CAT in our previous studies, which were increased at 7d, 15d and 30d. under NaCl stress, compared with the control (Y Chen et al. 2021, 49(15): 142-146, Jiangsu Agricultural Science, in Chinese), Here, we supplied the enzymatic activities of SOD, POD and CAT in our previous studies, which were increased at 2d (48h) and 7d (168h) in the newly revised manuscript (as shown in Fig. 1). 

Figure 1

(Note: The different letters indicate significant differences at the P < 0.05 level

The graph illustrates the changes of enzyme activities in SOD, POD and CAT compared with CK group at 48h and 168h after 200mM NaCl treatment.)

6. 3.7 The title is inappropriate. According to what the 8 genes were concerned and screened from all DEGs? Only these eight? Are there other TFs involved in stress regulation.

Response: According to the transcriptome sequencing of T. ramosissima leaves, many transcription factors were discovered. In this study, we did select 8 statically significant DEGs, which were specifically observed and annotated in the KEGG database.

7. 3.8 Unigenes mentioned in 3.6 and 3.7 should be selected and analyzed by qRT-PCR.

Response: Thank you for your valuable comments. In order to verify the reliability of the transcriptome sequencing data, we randomly selected 8 DEGs (including some of the transcription factors screened in this article), according to the description of verification methods in previous publications (He et al., Forests. 2020, 11(8); Lu et al., Biotechnology Bulletin (in Chinese).2020, 36(12):42-53).

.

8 The authors should detect the related enzyme activities such as POD, SOD, CAT etc. from control and treated T. ramosissima to support the conclusion.

Response: Thank you for your kind suggestions. Definitely, we carried out such experiments previously as mentioned above (Chen et al. 2021, 49(15): 142-146, Jiangsu Agricultural Science, in Chinese).

Discussion：

1. The author should condense the discussion part.

Response: We tried our best to rewrite the discussion part. Please see the newly submitted manuscript.

2. Line 284: 4.1 The results are not enough to support the conclusion. 

Response: Thanks for the kind reminder. After second thoughts, we revised the content as follows: Enhanced active oxygen scavenging capacity in T. ramosissima.

3. Line 290: “Overexpression of SbGST in transgenic tobacco plays a crucial role in salt stress tolerance” What is the role?

Response: These findings in tobacco showed that the expression of SbGST gene was up-regulated under the stress conditions of salt, cold, drought and salicylic acid. Among them, overexpression of SbGST gene in transgenic tobacco promoted seed germination and growth under salt stress. These results confirmed that the expression of SbGST genes was up-regulated under different stresses, and the overexpression of tua class SbGST genes in transgenic tobacco played a crucial role in abiotic stress tolerance. We revised the manuscript in the revised manuscript.

4. Lines 310-313 :4.2What do the authors want to say?

Response: In the 4.2 section of the Discussion part, we want to discuss the differentially expressed responsive transcription factor genes and their possible role under NaCl treatment ins T. ramosissima. 

5. Lines 322-323 : What led the authors to this conclusion about “that this gene needs to be active especially in a specific period under salt stress”? The results are not enough to support the conclusion.

Response: Many thanks for your valuable comments. After second thought, we have already improved this part in the newly submitted manuscript. 

Others：

1. Where's the legends of the figures？

Response: Sorry for the careless mistakes. The legends of the figures are supplied in the revised manuscript. Thanks a lot.

2. Writing problems in L149, 152, 160, 210,223,225,255,264,271.

Response: Sorry for our spelling mistakes. All problems have been revised in the newly submitted manuscript.

 

To Reviewer 2

Reviewer #2: A transcriptome analysis of T. ramosissima in response to NaCl stress was performed in this study, and some differentially expressed genes (DEGs) were found. However, the methods, results and conclusions should be considered more appropriately and rigorously.

Response: 

Response: Appreciate your valuable comments and suggestions. We carefully revised the manuscript and tried our best to improve it to be a better state to meet the journal’s requirements. The revised manuscript has been submitted to the journal, and looking forward to your further consideration

Introduction Section:

1. Lines 65-66: GhWRKY34 is not a wheat gene, please correct it after carefully reading the reference 20.

Response: Thank you so much, and sorry for our typo. GhWRKY34 is a gene belonging to cotton, which has been revised in the new submission. 

Materials and Methods Section:

1. In terms of the materials, why choose leaves not roots to study molecular mechanisms towards salt stress?

Response：Thank you for your valuable comments. Frankly, the root growth of T. ramosissima is quite slow and rare, especially under salt stress, and the phenotype of roots is not easily altered under the short term of NaCl treatment. While the leaves are the dominant fresh biomass in T. ramosissima, and exhibited salt secretion even under the short term of salt stress, which was not observed in roots (Fig. 2). Therefore, we chose the leaves of T. ramosissima as the research object to explore responsive DEGs under salt stress.

2. Lines 109-110: The filtering parameters of Fastp should be added. The original data was first filtered to obtain clean reads, then assembled! Not “the original data was filtered and assembled to obtain clean reads.” Please correct it, and add the method as well as software used in assembly.

Response: Thanks a lot for your valuable suggestion. The filtering parameters of Fastp were set in accordance with Chen et al. (2018, Bioinformatics, 34(17):i884-i890), which has been revised in the manuscript.

Results section:

Table 1: Please use scientific notation for the data.

Response: According to your valuable suggestion, it has been revised throughout the manuscript. 

2. Line 170: “We speculate that T. ramosissima leaves may respond to high NaCl stress by inhibiting the expression level of related DEGs at the transcription level.” What exactly does these down-regulated “DEGs” related to?

Response：Thanks for the useful comment. After second thought, we delete this sentence to avoid misunderstanding. 

3. Lines 201-202: “Results showed the expression of 39 ROS scavenging-related DEGs in the leaves of T.ramosissima were changed under salt treatment.” Please revise it.

Response: According to your suggestion, this sentence was changed into: ‘Results showed that the 39 ROS scavenging-related DEGs were responsive to salt treatment in the leaves of T. ramosissima’.

4. Lines 223-224: How many transcription factors in DEGs? Why only WRKY and bZIP were mentioned? What’s the P value and log2FC of these 8 genes?

Response: In total, more than 300 differentially expressed transcription factors genes were screened in this study partially annotated in GO and KEGG databases. In order to verify the reliability of the transcriptome sequencing data, we randomly selected 8 DEGs, which were specifically observed and annotated in the KEGG database, in this study. In addition, the P values of these 8 differentially expressed transcription factor genes were all 0 among the comparisons of CK-0h v.s. NaCl-48h, NaCl-48h vs NaCl-168h and CK-0h v.s. NaCl-168h, indicating a significant high Log2 Fold Change of 8 differentially expressed transcription factors. All these details mentioned here were listed in the new Table 1. Appreciate your valuable comments.

Table 1

Gene ID Description Pathway Log2 Fold Change

 CK-0h v.s. NaCl-48h NaCl-48h v.s. NaCl-168h CK-0h v.s. NaCl-168h

Unigene0010090 Transcription factor WRKY33 ko04626;ko04016 -0.75 0.35 -0.40

Unigene0014406 WRKY DNA-binding protein 27 ko04626;ko04016 0.46 0.33 0.79

Unigene0024962 WRKY transcription factor 1 ko04626 0.64 0.05 0.69

Unigene0077293 WRKY transcription factor ko04626;ko04016 -1.96 -0.08 -2.04

Unigene0079542 WRKY transcription factor 11 ko04626 -0.20. 0.38 0.18

Unigene0026888 bZIP4 ko04016 0.67 -0.31 0.36

Unigene0008868 bZIP2 ko01100;ko01110;ko01200;ko01212;ko04146;ko00071;ko00640;ko00410;ko01040;ko00592 0.64 -0.65 -0.01

Unigene0010561 bZIP10 ko04075 -0.82 0.43 -0.39

5. Lines 237-238: What kind of genes are Unigene0104732, Unigene0083695 and Unigene0069097? Why randomly choose these three genes instead of choosing genes mentioned above, like WKRY, bZIP?

Response: More than 300 differentially expressed transcription factors genes were screened in this study partially annotated in GO and KEGG databases. In order to verify the reliability of the transcriptome sequencing data, we randomly selected 8 DEGs, which were specifically observed and annotated in the KEGG database, according to the description of verification methods in previous publications (He et al., Forests. 2020, 11(8); Lu et al., Biotechnology Bulletin (in Chinese).2020, 36(12):42-53).

6. One of the main conclusion, “T. ramosissima may adapt to salt stress by enhancing the ROS scavenging mechanism, including SOD, POD, CAT, APX, GPX, GST, and GR.” I don’t think this conclusion is reliable. As the result showed there were some SOD, POD and CAT down-regulated whether after 48 h or 168 h salt treatment. The enzyme activity of SOD, POD, CAT should be measured to see whether ROS scavenging mechanism is enhanced.

Response: Frankly speaking, we did analyze the enzymatic activities of SOD, POD and CAT in our previous studies, which were increased at 7d, 15d and 30d. under NaCl stress, compared with the control (Y Chen et al. 2021, 49(15): 142-146, Jiangsu Agricultural Science, in Chinese), Here, we supplied the enzymatic activities of SOD, POD and CAT in our previous studies, which were increased at 2d (48h) and 7d (168h) in the newly revised manuscript (as shown in Fig. 1).

Figure 1

(Note: The different letters indicate significant differences at the P < 0.05 level

The graph illustrates the changes of enzyme activities in SOD, POD and CAT compared with CK group at 48h and 168h after 200mM NaCl treatment.)

7. Besides, the article needs careful revision; there are many typos and format error in it.

Response: We would like to thank you again for these valuable comments. We carefully revised the manuscript and tried our best to improve the English level. Hopefully, this revision work will meet your requirement! We need your further help and consideration. Many thanks again！

---

## [Decision Letter · Decision Letter 1]

28 Feb 2022

PONE-D-21-38993R1Transcriptome analysis of Tamarix ramosissima leaves in response to NaCl stressPLOS ONE

Dear Dr. Zhang,

Thank you for submitting your manuscript to PLOS ONE. After careful consideration, we feel that it has merit but does not fully meet PLOS ONE’s publication criteria as it currently stands.  There are still some minor amendments to be made to improve the paper quality. Therefore, we invite you to submit a revised version of the manuscript that addresses the points raised by the reviewers.

We look forward to receiving your revised manuscript.

Kind regards,

Guangxiao Yang, Ph.D

Academic Editor

PLOS ONE

Journal Requirements:

Reviewers' comments:

Reviewer's Responses to Questions

**Comments to the Author**

1. If the authors have adequately addressed your comments raised in a previous round of review and you feel that this manuscript is now acceptable for publication, you may indicate that here to bypass the “Comments to the Author” section, enter your conflict of interest statement in the “Confidential to Editor” section, and submit your "Accept" recommendation.

Reviewer #1: (No Response)

Reviewer #2: (No Response)

2. Is the manuscript technically sound, and do the data support the conclusions?

Reviewer #1: Yes

Reviewer #2: Partly

3. Has the statistical analysis been performed appropriately and rigorously? 

Reviewer #1: Yes

Reviewer #2: Yes

4. Have the authors made all data underlying the findings in their manuscript fully available?

Reviewer #1: Yes

Reviewer #2: Yes

5. Is the manuscript presented in an intelligible fashion and written in standard English?

Reviewer #1: No

Reviewer #2: Yes

6. Review Comments to the Author

Reviewer #1: The authors have revised some contents. There are still some details that need to be carefully revised. Please revise the manuscript carefully.

For example:

1. L31-32, L37-38. Please restate the relevant contents of transcription factors

2. L36 “the enzyme activities of SOD, POD, and CAT were significantly enhanced under NaCl treatment”. The CAT activity had no significant change in Figure2.

3.L88 “NaCl stress and screened and verified EGs at the transcriptional level.” Please rewrite this sentence

4.L99 “maintained at 26 ± 2 ℃ (day)() with a relative humidity of 40” Please rewrite this sentence

5. L125 “2.5 Differential gene screening” Please rewrite this little title

6. L127: maybe “corrected P value” and “FDR” is the same parameters?

7. L162: “Figure.2 Changes of SOD, Pod and CAT enzyme”, Pod or POD?

8. L209-210: “We speculate that T. ramosissima leaves may respond to high NaCl stress by reducing the expression level of related DEGs at the transcription level.” This conclusion is inappropriate.

9.L228 “3.6 KEGG pathway analysis of DEGs”

L275 “3.8 KEGG pathway analysis of key DEGs” Please rewrite these little titles

10. The reason why or how the author chose to pay attention to the eight transcription factors should be explained clearly in the manuscript

……

And the language of the manuscript needs to be carefully revised.

Please revise the manuscript carefully.

Reviewer #2: Lines 70-71: “In cotton, the WRKY transcription factor gene GhWRKY34 was induced by salt stress in transgenic Arabidopsis and wheat.” This sentence is still inaccurate, please correct it after carefully reading the reference.

Figure 1 : The lower left corner of the Figure 1 was CK-48h? not NaCl-0h? Besides, please add scale bar.

Figure 2: Please make sure that the data of Figure 2 is unpublished in your previous studies.

Table 1: "0.00%" ? Besides, please use scientific notation for the data.

Lines 263-265: “However, the antioxidant mechanism might be initially inhibited in T. ramosissima leaves under a long time (168 h) of NaCl treatment, just to adapt to salt stress.” How do you explain the fact that the antioxidant mechanism was significantly enhanced after 168 h of NaCl treatment (Figure 2).

As many DEGs were mentioned in 3.7 and 3.8, and some conclusions had been drawn, at least 2-3 genes of the these DEGs should be selected and analyzed by qRT-PCR to support your results and conclusion.

P in “P value” should be italic.

All the Figure legends and Table titles should be corrected to meet PLOS ONE's requirements.

7. PLOS authors have the option to publish the peer review history of their article (what does this mean?). If published, this will include your full peer review and any attached files.

Reviewer #1: No

Reviewer #2: No

---

## [Author Response · Author response to Decision Letter 1]

3 Mar 2022

Journal: PLOS ONE

Title: Transcriptome analysis of Tamarix ramosissima leaves in response to NaCl stress

Manuscript ID: PONE-D-21-38993

Dear Edit, 

Thank you for giving us the opportunity to resubmit our revised paper for publication in Plos One. We greatly appreciate the efforts of you, the associate editor and all reviewers for the constructive comments that have helped shape this manuscript into a better form. We have thoroughly revised the article and addressed all the concerns by either editing the manuscript or clarifying the details in the revised manuscript. Please see the detailed point-by-point response (marked in blue) added below. A 'point-by-point response' to each of the reviewer comments and a marked-up version of the manuscript with the changes highlighted in blue were also uploaded with the clean version of the manuscript files.

Sincerely,

Ning Zhang, Ph.D.

Faculty of Forestry, The University of British Columbia, 

Vancouver, British Columbia, Canada 

 

Response to Editor

1. Please upload a Response to Reviewers letter which should include a point by point response to each of the points made by the Editor and / or Reviewers. (This should be uploaded as a 'Response to Reviewers' file type.)

Response: Thank you for your valuable comments. We have finished editing

2. Please remove your figures/ from within your manuscript file, leaving only the individual TIFF/EPS image files. These will be automatically included in the reviewer’s PDF.

Response: According to your suggestion, We have removed all pictures in the manuscript from the manuscript, and each picture is packaged separately as a PDF file for upload.

 

Response to Reviewer 1

Dear Reviewer:

Many thanks for all your valuable comments and suggestions. We carefully revised the manuscript and tried our best to improve it to be a better state to meet the journal’s requirements. The revised manuscript has been submitted to the journal, and looking forward to your further consideration. 

Abstract 

1. Lines 31-32, Lines 37-38. Please restate the relevant contents of transcription factors

Response: According to your suggestion, we restated all these relevant contents of transcription factors in the revised manuscript. Thank you so much for valuable comments.

2. Line 36 “the enzyme activities of SOD, POD, and CAT were significantly enhanced under NaCl treatment”. The CAT activity had no significant change in Figure2.

Response: Thank you for your useful suggestions. We improved the description of this sentence in the revised manuscript that as following: ‘The enzyme activities of SOD and POD were significantly enhanced under NaCl treatment, but the enzyme activity of CAT was not significantly enhanced’.

3. Line 88 “NaCl stress and screened and verified EGs at the transcriptional level.” Please rewrite this sentence

Response: Sorry for our mistake. In the revised manuscript, it was revised as: ‘NaCl stress and screened and verified DEGs at the transcriptional level’.

Method：

1. Line 99 “maintained at 26 ± 2 ℃ (day)() with a relative humidity of 40” Please rewrite this sentence.

Response: Sorry for our mistake. This sentence was revised in the newly submitted manuscript that followed as: ‘maintained at 26 ± 2 ℃ (day) whose relative humidity stays between 40% and 55%’.

2. Line 125 “2.5 Differential gene screening” Please rewrite this little title

Response: According to your valuable suggestion, it has been revised throughout the manuscript as follows: ‘2.5 Screening methods for differentially expressed genes’.

Results:

1. Line 127: maybe “corrected P value” and “FDR” is the same parameters?

Response: Many thanks for valuable comments. FDR value means BH corrected p value. This has been revised in the manuscript. 

2. Lines 209-210: “We speculate that T. ramosissima leaves may respond to high NaCl stress by reducing the expression level of related DEGs at the transcription level.” This conclusion is inappropriate.

Response: Thank you for your valuable comments. It has been revised in the manuscript that followed as: ‘We speculate that T. ramosissima leaves may respond to high NaCl stress by affecting the expression level of related DEGs at the transcription level’.

3. Line 228 “3.6 KEGG pathway analysis of DEGs”. Line 275 “3.8 KEGG pathway analysis of key DEGs”. Please rewrite these little titles.

Reply: We rectified these little titles in the newly submitted manuscript as follows: ‘KEGG pathway analysis of DEGs in T. ramosissima leaves under NaCl stress’ and ‘3.8 Analysis of transcription factor DEGs in T. ramosissima leaves under NaCl stress’.

Others：

1. The reason why or how the author chose to pay attention to the eight transcription factors should be explained clearly in the manuscript.

Response: Many thanks for your valuable comments. It has been revised in the manuscript.

2. And the language of the manuscript needs to be carefully revised. Please revise the manuscript carefully.

Response: We would like to thank you again for these valuable comments. We carefully revised the manuscript and tried our best to improve the English level. Hopefully, this revision work will meet your requirement! We need your further help and consideration. Many thanks again!

 

To Reviewer 2

Dear Reviewer:

Response: Appreciate your valuable comments and suggestions. We carefully revised the manuscript and tried our best to improve it to be a better state to meet the journal’s requirements. The revised manuscript has been submitted to the journal, and looking forward to your further consideration

Introduction Section:

1. Lines 70-71: “In cotton, the WRKY transcription factor gene GhWRKY34 was induced by salt stress in transgenic Arabidopsis and wheat.” This sentence is still inaccurate, please correct it after carefully reading the reference.

Response: Sorry for our mistakes. It has been revised in the manuscript that followed as: ‘In cotton, the WRKY transcription factor gene GhWRKY34 was induced by salt stress in transgenic Arabidopsis’.

Results:

1. Figure 1：The lower left corner of the Figure 1 was CK-48h? not NaCl-0h? Besides, please add scale bar.

Response: Sorry for our mistake. It has been revised in the manuscript.

2. Figure 2: Please make sure that the data of Figure 2 is unpublished in your previous studies.

Response: We indeed confirm that the data for Figure 2 is not published previously.

3. Table 1: "0.00%" ? Besides, please use scientific notation for the data.

Response: Sorry for our mistake. It has been deleted in the manuscript.

4. Lines 263-265: “However, the antioxidant mechanism might be initially inhibited in T. ramosissima leaves under a long time (168 h) of NaCl treatment, just to adapt to salt stress.” How do you explain the fact that the antioxidant mechanism was significantly enhanced after 168 h of NaCl treatment (Figure 2).

Response: Sorry for the inaccurate description appeared in the original submission. Under prolonged (168 hours) NaCl treatment, the antioxidant machinery of T. ramosissima leaves may play a key role in up-regulating genes to adapt to salt stress.

5. As many DEGs were mentioned in 3.7 and 3.8, and some conclusions had been drawn, at least 2-3 genes of the these DEGs should be selected and analyzed by qRT-PCR to support your results and conclusion.

Response: Thank you for your valuable comments. In order to verify the reliability of the transcriptome sequencing data, we randomly selected 8 DEGs (including some of the transcription factors screened in this article), according to the description of verification methods in previous publications [He et al., Forests. 2020, 11(8); Lu et al., Biotechnology Bulletin (in Chinese).2020, 36(12):42-53]. In addition, Unigene0024962, as a verified gene, is also a transcription factor annotated to the KEGG Pathway mentioned above.

6. P in “P value” should be italic. All the Figure legends and Table titles should be corrected to meet PLOS ONE's requirements.

Response：According to your valuable suggestion, “p value” was written in italic, and all Figure legends and Table titles was corrected in the newly submitted manuscript.

We would like to thank you again for these valuable comments. We carefully revised the manuscript and tried our best to improve the English level. Hopefully, this revision work will meet your requirement! We need your further help and consideration.

---

## [Editor Report · Decision Letter 2]

7 Mar 2022

Transcriptome analysis of Tamarix ramosissima leaves in response to NaCl stress

PONE-D-21-38993R2

Dear Dr. Zhang,

We’re pleased to inform you that your manuscript has been judged scientifically suitable for publication and will be formally accepted for publication once it meets all outstanding technical requirements.

Kind regards,

Guangxiao Yang, Ph.D

Academic Editor

PLOS ONE
---

## [Editor Report · Acceptance letter]

21 Mar 2022

PONE-D-21-38993R2 

Transcriptome analysis of *Tamarix ramosissima* leaves in response to NaCl stress 

Dear Dr. Zhang:

I'm pleased to inform you that your manuscript has been deemed suitable for publication in PLOS ONE. Congratulations! Your manuscript is now with our production department. 

Kind regards, 

on behalf of

Dr Guangxiao Yang 

Academic Editor

PLOS ONE